# Effect of intra-dialytic pedaling exercise on dialysis adequacy: A randomized controlled trial

**Mahmoud Mohamadizadeh[1], Sharif Sharifi[2], Niloufar Motamed[3], Mohammad Amin Shadman[4], Shahnaz Pouladi** [ID][5]*

**1** Student Research Committee, Bushehr University of Medical Sciences, Bushehr, Islamic Republic of Iran, **2** Nursing and Midwifery Faculty of Bushehr University of Medical Sciences, Bushehr, Islamic Republic of Iran, **3** Department of Community Medicine, Medicine faculty of Bushehr University of Medical Sciences, Bushehr, Islamic Republic of Iran, **4** Nursing and Midwifery Faculty of Bushehr University of Medical Sciences, Bushehr, Islamic Republic of Iran, **5** Nursing and Midwifery Faculty of Bushehr University of Medical Sciences, Bushehr, Islamic Republic of Iran

* pouladi2008@gmail.com

## Abstract

### Background

In patients with chronic kidney disease undergoing hemodialysis, physical activity and rehabilitation are crucial for preventing declines in muscle strength and functional capacity. This study aimed to assess the impact of physical activity during hemodialysis on dialysis adequacy in patients undergoing hemodialysis.

### Methods

This randomized controlled trial (RCT) investigated the impact of pedaling exercise on dialysis effectiveness in 84 hemodialysis patients at hospitals in Bushehr. Participants were randomly assigned to either an experimental group (n = 42) that performed 30 minutes of pedaling exercise during their 4-hour dialysis sessions or a control group (n = 42) that received routine hemodialysis. Dialysis adequacy was assessed by comparing pre- and post-dialysis blood samples obtained from the arterial line. A conservative intradialytic exercise protocol, blood samples, and patient weight were measured using a calibrated digital scale. Data analysis was performed using SPSS version 24 software.

### Results

The experimental and control groups were similar in demographic characteristics, except for age (X2 = −3.84, p = 0.001) and education levels (X2 = 10.100, p = 0.006). While there was no significant difference in weight between the groups before and after the intervention (t = 0.223, p = 0.82 before; t = 0.203, p = 0.84 after), both groups showed a substantial weight reduction overall (p < 0.0001). There were no statistically

provided the original author and source are credited.

**Data availability statement:** The minimal data set underlying the findings of this study—including the raw values used to generate tables, figures, and statistical analyses—is provided in Supporting Information file S1_Data (SPSS format). These data are fully anonymized and sufficient to replicate the study's results.

**Funding:** The Bushehr University of Medical Sciences provided financial support for this study in the form of a Vice Chancellery of Research and Technology grant awarded to SP (IR.BPUMS.REC.1398.130). No additional external funding was received for this study. The funders had no role in study design, data collection and analysis, decision to publish, or preparation of the manuscript.

**Competing interests:** The authors have declared that no competing interests exist.

significant differences in weight change (t = 0.80, p = 0.25), BUN (t = 0.13, p = 1.52), or Kt/V (t = 1.62, p = 0.11) between the experimental and control groups.

## Conclusion

This study found that incorporating pedaling exercise during hemodialysis did not significantly improve dialysis effectiveness, as measured by weight change, BUN levels, or Kt/V. While both groups showed weight loss, there were no statistically significant differences between them. However, the study was limited by its small sample size and the specific exercise protocol employed. Further research with larger cohorts and a broader range of physical activities is needed to determine whether physical activity during hemodialysis can improve dialysis adequacy and overall patient outcomes.

## Trial registration

IRCT code number 20150529022466N15 and trial Code of Ethics IR.BPUMS. REC.1398.130E.

## Introduction

Chronic kidney disease (CKD) is a significant global health problem, leading to both short- and long-term complications and reducing patients' quality of life [1,2]. Recent studies indicate that CKD prevalence varies by region [3,4], with estimates of 11.4% in Iran [5], 9% in Mexico [6], and 13.1% among adults in the United States [7]. Several treatment options are available for managing CKD, including kidney transplantation, peritoneal dialysis, and hemodialysis [8]. Nearly 4 million people worldwide rely on kidney replacement therapy (KRT), with hemodialysis being the most common, accounting for about 69% of all KRT and 89% of dialysis cases [9]. CKD is projected to become the fifth leading cause of death globally by 2040 [10]. Given population growth, an annual increase of 5–6% in CKD prevalence poses a challenge for all countries worldwide. In Iran, this growth rate surpasses the global average [11].

Hemodialysis remains the most widely utilized modality for KRT worldwide, accounting for the majority of dialysis cases and imposing substantial financial and operational burdens on healthcare systems. Recent global epidemiological data confirm this trend, highlighting the increasing reliance on hemodialysis across diverse regions and its implications for public health planning [12].

Different hemodialysis programs exist, including home hemodialysis (3–6 times per week), in-center hemodialysis, short daily hemodialysis, and long nocturnal hemodialysis [13]. Although hemodialysis prevents death due to uremia, the survival rate of patients with CKD is significantly lower than that of the general population [14–16]. Several factors influence the survival rate of these patients, including the cause of renal failure, the treatment method, the presence of comorbidities such as cardiovascular disease, diabetes, and hypertension, and the adequacy of dialysis [17,18].

The long-term prognosis of patients undergoing dialysis depends heavily on the efficacy of their treatment, commonly referred to as dialysis adequacy. Defined by the National Kidney Foundation Kidney Disease Outcomes Quality Initiative (NKF KDOQI) as sufficient urea clearance, dialysis adequacy serves as a key indicator of treatment success and patient survival [19,20]. Inadequate dialysis adequacy has been associated with a range of adverse outcomes, including sexual dysfunction, cardiovascular disease, pruritus, prolonged hospitalization, and increased healthcare costs [21,22]. Conversely, improved dialysis adequacy enhances patients' quality of life, physical functioning, and spiritual well-being [23]. Given its central role in clinical outcomes, dialysis adequacy is not only a benchmark for evaluating dialysis protocols but also a critical endpoint in studies assessing adjunctive interventions such as intradialytic exercise. Understanding how physical activity during dialysis may influence adequacy is therefore essential to optimizing care strategies for this population.

The effectiveness of a dialysis session is commonly assessed using two key indicators: Kt/V and the urea reduction ratio (URR) [24,25]. Kt/V quantifies the efficiency of dialysis in removing urea from the body. It is calculated by dividing the product of dialyzer clearance (K) and dialysis time (t) by the urea volume of distribution, which approximates total body water (V) [26]. According to the urea kinetic model, both Kt/V and URR are standard measures of dialysis adequacy. Clinically, a minimum Kt/V value of 1.2 and a target URR of approximately 65% are recommended to ensure sufficient urea clearance during treatment [20].

Patients with CKD undergoing hemodialysis often experience functional decline, reduced quality of life, and psychological disorders such as depression and anxiety related to their treatment [27–30]. Hemodialysis is typically administered in a supine position, requiring patients to remain physically inactive for approximately 12 hours per week—equivalent to nearly 800 hours annually. This prolonged sedentary time contributes to complications such as reduced physical endurance, muscle atrophy, and muscle weakness [31]. Incorporating regular exercise during dialysis sessions has been shown to improve physical performance and mitigate muscle deterioration in hemodialysis patients [32,33].

Cycling represents a practical and safe intervention modality well suited to the physical constraints of hemodialysis patients [34]. Stationary pedaling can be performed during dialysis sessions without disrupting treatment, and recent evidence indicates that intradialytic cycling is well tolerated and easily supervised, thereby enhancing adherence and minimizing risk [34–36]. Moreover, it provides a unique opportunity to transform otherwise sedentary treatment time into therapeutic benefit.

Previous studies on intradialytic exercise have reported inconsistent findings, which appear to be influenced by differences in exercise volume and duration. Short-term or low-intensity interventions, such as those by Hatef et al. (2017), Kirkman et al. (2019), and Riahi (2012), did not demonstrate significant improvements in dialysis adequacy [37–39]. In contrast, longer-term or more structured programs, including combined aerobic and resistance training or higher-intensity protocols, have shown favorable effects on Kt/V and related clinical outcomes [40–42]. These discrepancies underscore the importance of standardized reporting of exercise volume across all groups to ensure transparency and comparability, and they highlight the need for further context-sensitive investigations.

Bushehr, located in southern Iran, faces unique environmental challenges; its hot, humid climate limits outdoor physical activity, particularly for patients with chronic conditions [43]. Moreover, the province has a disproportionately high number of hemodialysis patients per capita. Given the central role of dialysis nurses in delivering integrated care and patient education, their involvement presents a valuable opportunity to implement structured intradialytic exercise programs [34,35]. Therefore, this study was conducted to evaluate the effectiveness of supervised physical activity during hemodialysis on dialysis adequacy among patients treated at clinical centers in Bushehr, addressing both regional constraints and methodological gaps in prior research.

## Methods

### Study design

This study was a randomized controlled clinical trial (RCT; IRCT20150529022466N15) conducted between January 10 and March 30, 2020, in the dialysis departments of Bushehr hospitals in southern Iran. The trial was designed and

reported in accordance with the CONSORT 2010 guidelines to ensure transparency and methodological rigor [44]. The authors confirm that all ongoing and related trials for this intervention are registered.

### Trial registration

This trial was prospectively registered on the Iranian Registry of Clinical Trials (IRCT) under identifier IRCT20150529022466N15 on 05 May 2020. The intervention commenced only after registration was completed. The study protocol received ethical approval (IR.BPUMS.REC.1398.130) from the Ethics Committee of Bushehr University of Medical Sciences on 22 December 2019. The authors confirm that all ongoing and related trials for this intervention are registered.

### Inclusion and Exclusion Criteria

Inclusion criteria were: age between 18 and 65 years, undergoing continuous hemodialysis treatment for at least 3 months without interruption, ability to walk independently for at least 2 minutes, no history of unstable angina, dyspnea, myocardial infarction within the past month, or congestive heart failure; no use of neuropsychiatric medications; tolerance of 4-hour dialysis sessions; and absence of hypokalemia (K<3.5 mEq/L). These criteria were defined based on clinical guidelines and prior trials in hemodialysis populations [45].

Exclusion criteria were applied only to individuals who initially met all inclusion criteria. These included: cardiovascular complications such as unstable angina, systolic blood pressure ≥160 mmHg or diastolic ≥120 mmHg, hypotension defined as systolic BP<90 mmHg or a drop >30 mmHg during dialysis, neurological, musculoskeletal, or vascular disorders (confirmed by physician), missing four or more dialysis sessions during the study period, kidney transplantation or death, vascular access problems, and unwillingness to continue participation [45].

### Sample Size and Sampling Method

Using G*Power 3.1.9.2, with a 95% confidence level and 80% power, the required sample size was 76 participants for an expected effect size (d=0.65). The mean and SD of dialysis adequacy (Kt/V) in the control group were 1.33 and 0.18, respectively; the expected mean in the intervention group was 1.45. To account for a 10% dropout rate, the sample size was increased to 84 (42 per group) using the formula:

$$n' = n/(1 - q).$$

A non-probabilistic convenience sampling method was used to recruit patients from hospitals in Bushehr. Although random allocation was applied, initial recruitment was based on availability and willingness, which may introduce selection bias [40].

### Randomization and Blinding

Participants were randomly assigned to intervention or control groups using a computer-generated random number table by an independent researcher. Allocation was concealed using sequentially numbered, sealed opaque envelopes. The study was single-blind: participants were unaware of their group assignment. Due to the nature of the intervention, blinding of medical staff was not feasible. To minimize contamination, groups were treated in separate centers and at different times [41].

### Treatment Uniformity and Vascular Access

All participants underwent conventional hemodialysis (not hemodiafiltration) three times per week for four hours per session, with identical dialysis modality and frequency across both study groups.

Detailed documentation of vascular access type (e.g., arteriovenous fistula, graft, or central venous catheter) was not included in the standardized nursing records at the participating centers during the study period. The clinical records for these prevalent hemodialysis patients primarily focused on session-specific parameters (dialysis duration, blood flow rate, ultrafiltration volume, and vital signs) and routine laboratory values. A retrospective review confirmed that access type was not consistently recorded for all participants, precluding its reliable extraction for formal baseline comparison or use as a covariate in the analysis. However, a post hoc assessment of available notes indicated that the vast majority of participants used arteriovenous fistulas, with no apparent systematic difference in access type between the study groups.

**Exercise Protocol and Adherence (CERT-Based Description)**

The intradialytic exercise intervention was designed and reported in accordance with the Consensus on Exercise Reporting Template (CERT) to ensure methodological transparency and replicability [46]. Participants in the intervention group performed stationary cycling during dialysis using a portable ATMED (Taiwan) bike weighing 3.5 kg, positioned at the foot of the dialysis bed. All exercises were conducted in the supine position during dialysis.

Each session consisted of a 5-minute warm-up, two 15-minute pedaling intervals separated by a 15-minute rest, and a cool-down phase. The intervention lasted 4 weeks, with three sessions per week, for a total of 12 sessions. Based on established exercise physiology guidelines, the intradialytic cycling intervention was classified as light-intensity physical activity. The exercise intensity was prescribed to elicit approximately a 10% increase in heart rate from baseline, which corresponds to a light workload in patients undergoing maintenance hemodialysis. Given the clinical context, participants were instructed to exercise at a comfortable pace without signs of exertional distress, consistent with light-intensity activity (i.e., below moderate-intensity thresholds). This conservative intensity was intentionally selected to ensure patient safety and hemodynamic stability during dialysis sessions. Exercise intensity was individualized and adjusted to achieve a 10% increase in heart rate compared to baseline. The safe pulse range during exercise was maintained at 50–125 beats per minute. Pulse and blood pressure were measured before and after each session, and pulse was continuously monitored throughout. If the target heart rate was not achieved, patients were instructed to pedal faster. [47].

All participants in the intervention group completed the full set of exercise sessions without interruption. Kt/V values were recorded during the 12-session intervention period, starting from the first dialysis session. As the exercise protocol commenced immediately, no pre-intervention Kt/V measurements were collected. This design reflects the pragmatic integration of the intervention into routine clinical practice [48].

All sessions were supervised by a trained nurse familiar with the protocol. Participants received verbal encouragement and feedback to maintain motivation and adherence. The exercise protocol was standardized across all participants, and no modifications were required during the study. The equipment and setting remained consistent throughout. The full protocol is archived in the thesis repository of Bushehr University of Medical Sciences and is available upon request [42].

Participants were monitored for signs of abnormality or adverse events during each exercise session. The supervising nurse documented clinical observations throughout the intervention period in accordance with the safety protocol.

**Data Collection Tools**

Data were collected using a demographic form and a dialysis adequacy form. Variables included sex, age, education, marital status, weight (pre- and post-dialysis), and dialysis adequacy indices. Kt/V was calculated using:

$$Kt/V = -\ln f_0(R - 0.008 \times t) + (4 - 3.5 \times R) \times UF/W$$

To calculate the Urea Reduction Ratio (URR), blood urea nitrogen (BUN) levels were measured before and after dialysis.

$$URR = (BUN\ pre - BUN\ post)/BUN\ pre$$

Blood samples were collected pre- and post-dialysis using standard protocols. Post-dialysis samples were collected after reducing the dialysate flow for 3 minutes and the pump speed to <100 cc/min for the final 5 minutes. Samples were coded and analyzed using a DIRUI 1200 machine. Weight was measured using a calibrated Seca digital scale [45].

## Implementation

After the study protocol was approved by the Ethics Committee of Bushehr University of Medical Sciences (IR.BPUMS. REC.1398.130), the researcher obtained an official letter of introduction and visited the dialysis departments of Bushehr Teaching Hospitals. Participant recruitment and the intervention were conducted between early June and late August 2020.

After coordinating with the clinical staff, the researcher introduced the study and its objectives to eligible patients, obtained written informed consent, and initiated sampling. Participants were assured that their data would remain confidential and that they could withdraw from the study at any time without any negative consequences.

A total of 84 patients undergoing hemodialysis were recruited and randomly assigned to either the intervention or control group. All patients received conventional hemodialysis three times per week for four hours per session [49]. The control group received standard care without any additional physical activity, remaining in the supine position throughout dialysis. The intervention group participated in intradialytic cycling using a portable ATMED (Taiwan) stationary bike weighing 3.5 kg, positioned at the foot of the dialysis bed.

Each exercise session consisted of two 15-minute pedaling intervals, separated by a 15-minute rest, preceded by a 5-minute warm-up and followed by a cool-down phase. The pedaling intensity was adjusted to achieve a 10% increase in heart rate from baseline. Blood pressure and pulse were measured before and after each session, and pulse was continuously monitored during exercise. If the target heart rate was not achieved, patients were instructed to pedal faster. The safe pulse range during exercise was maintained at 50–125 beats per minute.

The intervention program lasted four weeks, with three sessions per week, totaling 12 sessions [50,51]. All participants in the intervention group completed the full set of exercise sessions. Dialysis adequacy was measured before and after the intervention period using Kt/V and URR indices.

During the study, two participants from the intervention group and three from the control group withdrew due to fatigue or unwillingness to continue. These cases were excluded from the final analysis. No post-intervention follow-up period was included in this study. All outcome measurements, including dialysis adequacy (Kt/V), were collected immediately after the final session of the exercise protocol.

Ultimately, data from 40 participants in the intervention group and 39 in the control group were analyzed. The flow of participants through each stage of the study—including recruitment, randomization, follow-up, and analysis—is presented in the CONSORT flow diagram (Fig 1), providing a visual summary of trial conduct.

The entire trial protocol is archived in the library of Bushehr University of Medical Sciences and documented in the master's thesis of nursing student Mahmoud Mohammadizadeh.

## Ethical considerations

This study was part of a master's thesis in medical-surgical nursing at Bushehr University of Medical Sciences and registered under IRCT20150529022466N15. All participants received complete study information and signed informed consent. Confidentiality was maintained, and all procedures were conducted in accordance with the Declaration of Helsinki and relevant ethical guidelines.

## Data analysis

Following data collection, statistical analysis was conducted using SPSS version 24 (IBM Corp., Armonk, NY, USA). Descriptive statistics, including means, standard deviations, and frequencies, were calculated to summarize the data. Group comparisons of demographic variables were performed using chi-squared tests. Paired-samples t-tests were used

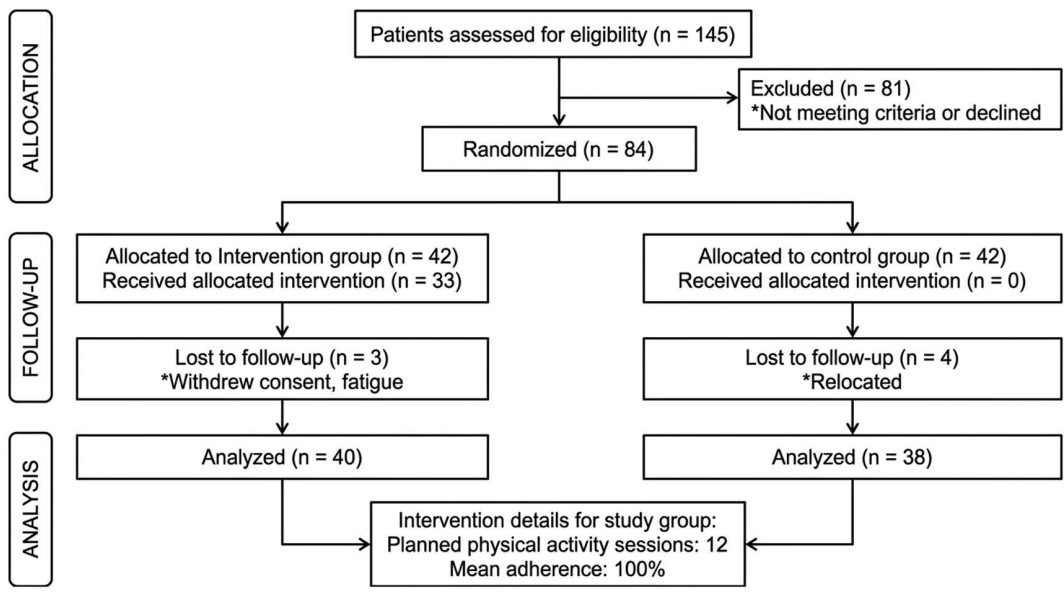

**Fig 1. Consort flow diagram.**

to assess within-group changes in dialysis adequacy (Kt/V) before and after the intervention, and independent-samples t-tests were used to compare the intervention and control groups.

Normality of continuous variables was assessed using the Shapiro-Wilk test and by evaluating the ratio of skewness to its standard error. A p-value of less than 0.05 was considered statistically significant.

Effect sizes (Cohen's d) and 95% confidence intervals were calculated for key outcomes to assess the magnitude and clinical relevance of the intervention. All analyses were performed using complete-case data, excluding participants who withdrew from the study.

Due to sample size limitations, no covariate-adjusted analysis was performed. All group comparisons were conducted using unadjusted statistical tests.

## Results

The present study aimed to evaluate the effect of a structured physical activity intervention on dialysis adequacy and selected biochemical and physiological parameters in patients undergoing hemodialysis. A total of 79 patients were enrolled, with 40 in the intervention group and 39 in the control group. All patients completed the study protocol and were included in the final analysis. No adverse events or clinical complications related to the exercise intervention were observed or reported during the study period. Five participants (two in the intervention group and three in the control group) withdrew from the study due to a general unwillingness to continue.

Demographic characteristics of the participants are presented in Table 1. The majority were male (n = 51, 64.6%) and married (n = 72, 91.1%). Regarding employment status, 33 patients (41.3%) were employed, and 50 (63.3%) had an education level below a diploma. A statistically significant difference in mean age was observed between the intervention and control groups (p = 0.001), with the control group having a higher mean age. No significant differences were found between the groups in terms of marital status (p = 0.11) or employment status (p = 0.29). A statistically significant difference was observed in education level (p = 0.006), with a higher proportion of patients in the control group having lower educational attainment.

**Table 1. Comparison of Demographic Variables Between the Intervention and Control Groups in Hospitals of Bushehr, Iran (2020).**

| Variable | Category | Intervention Group N (%) | Control Group N (%) | Total N (%) | X² | p-value | Effect Size (Cohen's d) and 95% CI |
|---|---|---|---|---|---|---|---|
| Gender | Male | 29 (72.5%) | 22 (56.4%) | 51 (64.6%) | 2.23 | 0.13 | |
| | Female | 11 (27.5%) | 17 (43.6%) | 28 (35.4%) | | | |
| Marital Status | Single | 6 (15%) | 1 (2.6%) | 7 (8.9%) | 3.78* | 0.11 | |
| | Married | 34 (85%) | 38 (97.4%) | 72 (91.1%) | | | |
| Employment Status | Unemployed | 15 (38.5%) | 19 (47.5%) | 34 (42.5%) | 2.42 | 0.29 | |
| | Employed | 16 (40%) | 17 (42.5%) | 33 (41.3%) | | | |
| | Retired | 9 (22.5%) | 4 (10%) | 13 (16.3%) | | | |
| Education Level | Below Diploma | 19 (47.5%) | 31 (79.5%) | 50 (63.3%) | 10.10 | 0.006 | |
| | Diploma | 10 (25%) | 6 (15.4%) | 16 (20.3%) | | | |
| | University Degree | 11 (27.5%) | 2 (5.1%) | 13 (16.5%) | | | |
| Age (years)** | — | 50.25±10.71 | 58.64±8.55 | — | −3.84 | 0.001 | d = −0.05, CI [−4.5, 3.3] |

*Fisher's exact test

** Mean±Standard Deviation

† Categorical variables were analyzed using the Chi-square test and Fisher's exact test. Continuous variable (age) was analyzed using an independent samples t-test. Effect sizes are reported as Cohen's d with 95% confidence intervals.

Table 2 summarizes the comparison of blood urea, body weight, and BUN before and after the intervention. No significant difference in blood urea levels was observed between the groups (p=0.13). Both groups showed a statistically significant reduction in body weight from pre- to post-intervention (p<0.0001), although the between-group differences in pre-intervention weight (p=0.82) and post-intervention weight (p=0.84) were not significant. Similarly, BUN levels decreased significantly within both groups (p=0.001), but no significant differences were found between the intervention and control groups at either time point (p=0.17 and p=0.74, respectively) (Fig 2).

Table 3 presents the mean changes in blood urea, body weight, and BUN between the two groups. None of these changes was statistically significant (p=0.13, 0.80, and 0.07, respectively), although the reduction in BUN approached the threshold for significance.

Table 4 shows the post-intervention Kt/V values, which serve as an indicator of dialysis adequacy. No statistically significant difference was found between the intervention and control groups (p=0.11) (Fig 3).

No adverse events or clinical complications related to the exercise intervention were reported during the 4-week study period.

## Discussion

### Effect of Intradialytic Physical Activity on Dialysis Adequacy (Primary Outcome)

The primary objective of the present study was to evaluate the effect of intradialytic physical activity on dialysis adequacy. The findings indicated that although the Kt/V ratio improved modestly in the intervention group following the structured physical activity program, this change was not statistically significant compared with the control group. From a physiological perspective, these findings should be interpreted in the context of the intervention's temporal nature, as the exercise protocol more closely resembles an acute than a chronic exercise paradigm.

In the present study, the intervention was implemented over four weeks and consisted of short-duration pedaling sessions performed during dialysis. While this design was intentionally selected to ensure feasibility, safety, and patient

**Table 2. Comparison of Blood Urea, Body Weight, and BUN Before and After the Intervention Between the Intervention and Control Groups in Patients Hospitalized in Bushehr, Iran (2020).**

| Variable | Time Point | Intervention Group (Mean±SD) | Control Group (Mean±SD) | t | p-value | Effect Size (Cohen's d) and 95% CI |
|---|---|---|---|---|---|---|
| Blood Urea Level | — | 0.616±0.049 | 0.601±0.040 | 1.52 | 0.13 | d=0.33, CI [−0.005, 0.035] |
| Body Weight | Before | 72.96±13.85 | 73.61±11.93 | −0.223 | 0.82 | |
| | After | 70.99±13.95 | 71.59±11.85 | −0.203 | 0.84 | |
| | t | 0.11 | 13.80 | | | d = −0.05, CI [−4.5, 3.3] |
| | p-value | 0.0001 | 0.0001 | | | |
| BUN | Before | 58.66±9.65 | 55.73±8.92 | 1.40 | 0.17 | – |
| | After | 22.53±4.69 | 22.18±4.51 | 0.34 | 0.74 | d=.08, CI [−1.3, 2.0] |
| | t | 34.08 | 37.49 | | | |
| | p-value | 0.001 | 0.001 | | | |

Independent samples t-tests were used to compare blood urea and BUN levels between groups at each time point. Paired-samples t-tests were used to assess within-group changes in body weight before and after the intervention. Effect sizes were calculated using Cohen's d with 95% confidence intervals.

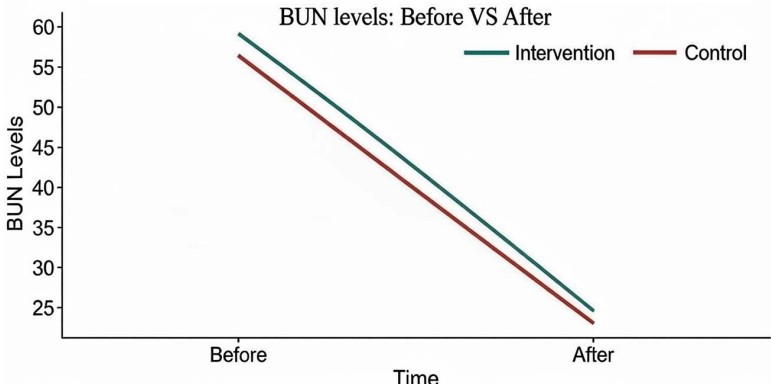

**Fig 2. Comparison of Mean Difference in Blood Urea (BUN) Levels Before and After Intervention in Intervention and Control Groups.**

**Table 3. Comparison of Mean Changes in Blood Urea, Body Weight, and BUN Between the Intervention and Control Groups in Patients Hospitalized in Bushehr, Iran (2020).**

| Variable | Intervention Group (Mean±SD) | Control Group (Mean±SD) | t | p-value | Effect Size (Cohen's d) and 95% CI |
|---|---|---|---|---|---|
| Blood Urea Change | 0.616±0.049 | 0.601±0.040 | 1.52 | 0.13 | d =.33, CI [−0.005, 0.035] |
| Weight Change | 1.96±1.13 | 2.02±0.91 | −0.25 | 0.80 | d = 0.06, CI [−0.45, 0.33] |
| BUN Change | 36.13±6.70 | 33.54±5.59 | 1.85 | 0.07 | d = 0.42, CI [−0.3, 5.5] |

Between-group comparisons were conducted using independent samples t-tests. Effect sizes were calculated using Cohen's d with 95% confidence intervals.

**Table 4. Comparison of Mean Kt/V Values After the Intervention Between the Intervention and Control Groups in Patients Hospitalized in Bushehr, Iran (2020).**

| Group | Intervention Group (Mean ± SD) | Control Group (Mean ± SD) | t | p-value | Effect Size (Cohen's d) and 95% CI |
|---|---|---|---|---|---|
| Mean Kt/V | 1.163 ± 0.14 | 1.115 ± 0.12 | 1.62 | 0.11 | d = 0.37, CI [−0.01, 0.11] |

A between-group comparison was performed using an independent-samples t-test. Effect size was calculated using Cohen's d with a 95% confidence interval.

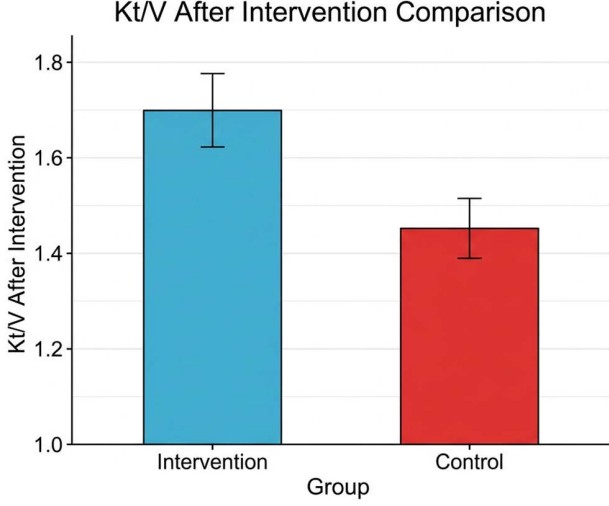

**Fig 3. Comparison of post-intervention Kt/V values between the intervention and control groups.**

tolerance, it is unlikely to have provided sufficient time for the development of chronic physiological adaptations, such as sustained improvements in muscle perfusion, metabolic efficiency, or solute transport capacity. In exercise physiology, meaningful and durable adaptations typically require prolonged and progressive exposure to training stimuli over several weeks to months. Therefore, the absence of statistically significant changes in dialysis adequacy may primarily reflect the insufficient duration of the intervention to induce chronic adaptations, rather than limitations related solely to exercise volume or intensity.

Beyond the temporal dimension, the intervention was also characterized by a conservative exercise regimen. The protocol consisted of 15-minute pedaling sessions performed three times per week and was delivered at a light-to-low–moderate intensity to minimize hemodynamic instability during dialysis. While clinically appropriate, this combination of short duration and low physiological stimulus further supports the interpretation of the intervention as predominantly acute. Acute bouts of exercise may transiently increase muscle blood flow and mobilize urea into the circulation [37,52]; however, without sufficient duration and repetition, these transient responses are unlikely to translate into sustained improvements in dialysis adequacy.

The lack of a significant effect on Kt/V is consistent with previous studies that employed short-term or low-intensity intradialytic exercise interventions. Hatef et al. (2017) [37], Kirkman et al. (2019) [38], and Riahi (2012) [39] similarly reported no significant improvements in dialysis adequacy, supporting the notion that acute or short-duration exercise interventions may be insufficient to elicit meaningful changes in solute clearance. In contrast, studies that reported

favorable effects on dialysis adequacy generally employed chronic, long-term exercise programs, often incorporating higher intensities, progressive overload, or multimodal training strategies. For example, Vaithilingam et al. (2004) demonstrated significant improvements in Kt/V following a prolonged combined aerobic and resistance training program in patients with stable vascular access [40]. Likewise, Pu et al. (2019) reported modest improvements in dialysis adequacy following a structured aerobic intervention with greater temporal consistency [41].

In addition to the acute nature of the intervention, unmeasured variability in vascular access function may have further contributed to heterogeneity in solute clearance. Although most participants utilized arteriovenous fistulas, access quality and flow characteristics were not systematically assessed. This methodological limitation, combined with the short intervention duration, may have attenuated the potential impact of intradialytic exercise on dialysis adequacy.

Taken together, these findings suggest that the present intervention primarily evaluated the short-term feasibility and acute physiological response to intradialytic physical activity rather than its capacity to induce chronic adaptations. The absence of statistically significant improvements in dialysis adequacy should therefore be interpreted as reflecting the intervention's acute nature and limited duration. Future studies should prioritize longer-term exercise programs with progressive training stimuli to better capture the potential chronic effects of intradialytic exercise on dialysis adequacy.

### Changes in Blood Urea Nitrogen and Body Weight

Although between-group comparisons did not reveal statistically significant differences, within the intervention group, BUN levels and body weight decreased. These changes may be consistent with transient physiological responses, such as increased muscle perfusion and mobilization of urea during physical activity [37,52,53]. The short duration of the intervention and the timing of dialysis may have limited the effective removal of mobilized solutes.

### Systematic evidence is consistent with these findings

A review by Kirkman et al. (2019) concluded that simple intradialytic exercise interventions, such as short-duration pedaling, do not consistently result in statistically significant improvements in dialysis adequacy indices [38]. In contrast, longer and more structured interventions have demonstrated statistically significant effects. For example, Pujiastuti et al. (2020) reported reductions in body weight and improvements in clinical parameters following a more prolonged structured exercise program [42]. Differences in intervention length, exercise intensity, and baseline patient characteristics may explain these discrepancies.

### Influence of Demographic and Methodological Factors

Baseline differences in age and educational level between the intervention and control groups represent an additional limitation that may have influenced the outcomes. Older patients may experience reduced physical capacity, slower physiological adaptation, and lower tolerance for exercise interventions [54]. Similarly, individuals with higher educational attainment often demonstrate greater health literacy, self-regulation, and adherence to structured health interventions [55]. These demographic disparities may have contributed to variability in exercise engagement and dialysis adequacy outcomes.

Furthermore, the relatively small sample size increases the risk of a Type II error, and the absence of covariate-adjusted analyses may have limited the study's ability to detect subtle intervention effects. Future studies should consider stratified randomization or statistical adjustment methods, such as ANCOVA, to control for baseline imbalances and enhance internal validity.

Another limitation of this study is the conservative exercise protocol intensity, which was intentionally selected to ensure patient safety during dialysis but may have reduced the measurable physiological and biochemical responses. This

conservative approach, while clinically appropriate in the intradialytic setting, may have reduced the intervention's capacity to elicit a sufficient metabolic and circulatory stimulus to produce measurable improvements in dialysis adequacy.

### Practical implications and Future Directions

Despite the lack of statistically significant improvements in dialysis adequacy, the intradialytic exercise program was feasible, well-tolerated, and safely implemented, with no adverse events. Even low-intensity physical activity during dialysis may offer indirect benefits, including improved circulation, enhanced patient engagement, and potential psychosocial advantages [54,55]. These findings support the clinical feasibility of integrating supervised intradialytic exercise into routine dialysis care.

Future research should focus on more extended intervention periods, higher exercise intensities, or multimodal exercise programs combining aerobic and resistance training. Incorporating standardized assessments of vascular access function and expanding outcome measures to include patient-reported and functional outcomes may further clarify the role of intradialytic exercise in optimizing dialysis care.

## Conclusion

In conclusion, this randomized controlled trial found that a 4-week intradialytic pedaling exercise intervention did not lead to a statistically significant improvement in the primary outcome of dialysis adequacy (Kt/V). Therefore, future research should prioritize a comprehensive re-evaluation of intervention parameters. Key considerations for subsequent investigations should include the careful selection of exercise modalities, optimization of session duration, and determination of the most efficacious timing of exercise relative to the hemodialysis procedure. Furthermore, rigorous control of confounding variables, such as patient age and educational attainment, is essential to minimize potential bias and ensure the validity of future findings. By systematically addressing these factors, future studies may be better positioned to identify effective strategies for integrating physical activity into the hemodialysis routine to improve patient outcomes.

Although no sensitivity analysis or age-matching was performed in the present study, the observed age differences between groups may have influenced the outcomes. Future trials should consider stratified randomization or statistical adjustments (e.g., ANCOVA) to account for age-related variability and enhance internal validity.

### Limitations

Several limitations should be considered when interpreting these findings, as they may affect both internal and external validity. First, an important methodological limitation of this study is the lack of systematically recorded baseline information on vascular access type. As detailed in the Methods, these data were not part of routine documentation, preventing their formal analysis. Although a post hoc review suggested no apparent imbalance in access type between groups and that arteriovenous fistulas were predominant, the absence of standardized data remains a limitation that could affect the interpretation of solute clearance outcomes. Future studies should prospectively document and analyze vascular access characteristics.

Second, the relatively small sample size (n = 79 after dropouts) may have limited the study's ability to detect minor yet potentially clinically meaningful differences between groups. Future studies with larger sample sizes are warranted to improve statistical power. Third, the intervention focused exclusively on low-intensity intradialytic pedaling exercise. The effects of other exercise modalities, such as resistance training, high-intensity aerobic exercise, or combined exercise programs, were not examined; therefore, the findings may not be generalizable to these interventions. Fourth, although a single-blind design was used, the lack of blinding of care providers and outcome assessors may have introduced performance or detection bias. Fifth, the study population consisted of hemodialysis patients from a single center in Bushehr, Iran, with specific inclusion criteria (ages 18–65 years and absence of severe comorbidities). Therefore, the generalizability of the findings to other populations, healthcare settings, or patients with more complex medical conditions may be

limited. Additionally, despite overall demographic similarity between groups, differences in age and education level could have introduced residual bias. Sixth, convenience sampling and single-center recruitment further limit the external validity of the findings. Seventh, the lack of pre-intervention Kt/V measurements restricted longitudinal assessment of dialysis adequacy. Because the intradialytic exercise protocol was initiated in the first dialysis session, obtaining a true baseline measurement was not feasible to maintain clinical practicality and minimize disruption to routine care. Finally, the study was conducted at a single center with the same treatment team supervising all sessions, which may limit the applicability of the findings to settings with different dialysis protocols, staffing patterns, or levels of clinical supervision.

## Supporting information

**S1 File. CONSORT 2010 Checklist.** This file contains the completed CONSORT checklist used to ensure transparent reporting of the randomized clinical trial.
(DOCX)

**S2 File. Persian Original Thesis.** This file contains the full Persian-language thesis of the Master of Nursing student whose research underpinned the current manuscript. It contains the complete trial protocol and data documentation.
(PDF)

**S3 File. Trial Protocol.** This document outlines the original protocol approved by the institutional ethics committee, including study design, intervention procedures, and outcome measures.
(DOCX)

**S4 File. Ethics Approval Certificate.** This file contains the official ethics approval letter issued by the Research Ethics Committee of Bushehr University of Medical Sciences for the submitted study.
(PDF)

**S5 File. SPSS dataset.** This file contains the fully anonymized SPSS dataset used for statistical analysis in the study. It is provided as Supporting Information and is publicly available to ensure transparency and reproducibility of the findings.
(SAV)

## Acknowledgments

This article is derived from a Master's thesis conducted with financial support from Bushehr University of Medical Sciences. We sincerely thank the hemodialysis patients at the hospitals' dialysis departments in Bushehr for their participation and cooperation.

## Author contributions

**Conceptualization:** Mahmoud Mohamadizadeh, Shahnaz Pouladi.

**Data curation:** Niloufar Motamed, Shahnaz Pouladi.

**Formal analysis:** Niloufar Motamed.

**Methodology:** Shahnaz Pouladi.

**Project administration:** Shahnaz Pouladi.

**Resources:** Mohammad Amin Shadman, Shahnaz Pouladi.

**Software:** Niloufar Motamed.

**Supervision:** Shahnaz Pouladi.

**Validation:** Shahnaz Pouladi.

**Visualization:** Sharif Sharifi.

**Writing – original draft:** Mahmoud Mohamadizadeh.

**Writing – review & editing:** Niloufar Motamed, Mohammad Amin Shadman, Shahnaz Pouladi.

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
