## [Decision Letter · Decision Letter 0]

12 Jun 2025

PONE-D-25-17351Effect of Intra-Dialytic Pedaling Exercise on Dialysis Adequacy: A Randomized Controlled TrialPLOS ONE

Dear Dr. Pouladi,

Thank you for submitting your manuscript to PLOS ONE. After careful consideration, we feel that it has merit but does not fully meet PLOS ONE’s publication criteria as it currently stands. Therefore, we invite you to submit a revised version of the manuscript that addresses the points raised during the review process.

We look forward to receiving your revised manuscript.

Kind regards,

Ankur Shah

Academic Editor

PLOS ONE

2. We note that you have selected “Clinical Trial” as your article type. PLOS ONE requires that all clinical trials are registered in an appropriate registry (the WHO list of approved registries is at "https://www.who.int/clinical-trials-registry-platform/network/primary-registries" https://www.who.int/clinical-trials-registry-platform/network/primary-registries and more information on trial registration is at http://www.icmje.org/about-icmje/faqs/clinical-trials-registration/). Please state the name of the registry and the registration number (e.g. ISRCTN or ClinicalTrials.gov) in the submission data and on the title page of your manuscript. a) Please provide the complete date range for participant recruitment and follow-up in the methods section of your manuscript. b) If you have not yet registered your trial in an appropriate registry, we now require you to do so and will need confirmation of the trial registry number before we can pass your paper to the next stage of review. Please include in the Methods section of your paper your reasons for not registering this study before enrolment of participants started. Please confirm that all related trials are registered by stating: “The authors confirm that all ongoing and related trials for this drug/intervention are registered”. Please see http://journals.plos.org/plosone/s/submission-guidelines#loc-clinical-trials for our policies on clinical trials.

4. In the online submission form, you indicated that ["The datasets generated and analyzed during the current study are available in the library of Bushehr University of Medical Sciences, in the thesis of a Master of Nursing student named Mahmoud Mohammadizadeh. The full trial protocol is also available in this thesis. Interested researchers may contact the corresponding author to request access to this data].

Additional Editor Comments:

The authors have performed a RCT evaluating the impact of intra-dialytic pedaling on dialysis adequacy. The reviewers highlight areas for improvement in clarity of the exposure and outcome as well as randomization.

Reviewers' comments:

Reviewer's Responses to Questions

**Comments to the Author**

1. Is the manuscript technically sound, and do the data support the conclusions?

Reviewer #1: Yes

Reviewer #2: Yes

Reviewer #3: Partly

Reviewer #4: Yes

2. Has the statistical analysis been performed appropriately and rigorously? 

Reviewer #1: Yes

Reviewer #2: Yes

Reviewer #3: Yes

Reviewer #4: No

3. Have the authors made all data underlying the findings in their manuscript fully available?

Reviewer #1: Yes

Reviewer #2: Yes

Reviewer #3: No

Reviewer #4: Yes

4. Is the manuscript presented in an intelligible fashion and written in standard English?

Reviewer #1: Yes

Reviewer #2: No

Reviewer #3: Yes

Reviewer #4: Yes

5. Review Comments to the Author

Reviewer #1: Authors recruited 84 hemodialysis patients with a randomized controlled trial to investigate the effectiveness of physical activity on dialysis effectiveness. The results no significant improvement in dialysis effectiveness with pedaling exercise during hemodialysis.

1. Line 179. The control group underwent hemodialysis using the routine method. Please clarify the “routine method”.

2. The experimental group and control group participated in different activities: Physical activity during dialysis vs. the routine method. Does the routine method include physical activity? If not, how to ensure to have a single blind study design?

3. Line 197. Some participants were excluded after the start of the study. Please clarify the exclusion criteria here.

4. Tables vs text. The decimal point was presented in different format which should be corrected to be consistent.

5. Line 215 mentioned ANOVA and Tukey’s post hoc test in the analysis section. Please clarify where the corresponding results for ANOVA/post hoc test were presented.

6. It seems that authors only conduct the comparison of two groups for before intervention and then another comparison of two groups for after intervention. As the baseline characteristics are not the same for some variables. It is unclear how these variables were taken into consideration in the analysis to evaluate the effective of physical activity.

Reviewer #2: Dear authors,

Congratulations on your study about exercise during hemodialysis and its effect on dialysis adequacy. The topic is important and relevant for improving the quality of life of patients with kidney disease. However, after careful review of the manuscript, I would like to highlight some points that need improvement to make the work clearer and more scientific.

First, it is essential that all data supporting the analyses and conclusions be available to readers, either as supplementary files or in a public repository. This allows other researchers to verify and use the results reliably. The statistical analysis is appropriate, but it would be helpful to include additional information such as effect sizes and confidence intervals, so readers better understand the clinical significance of the findings. I also suggest explaining how the sample size was calculated and how missing data or participant dropouts were handled.

Another important point is that there were significant differences between groups regarding age and education level, which might have influenced the results. This issue should be discussed more carefully, as it may affect the exercise impact. Additionally, the study was single-blind, meaning the assessors knew the group assignments, which could introduce some bias. This possible bias should be addressed in the manuscript.

The writing could be clearer and more concise, and I recommend a thorough English language review to correct any errors. The discussion section could also be more straightforward, avoiding repetition and better explaining the reasons behind the findings, especially the nonsignificant results. It is important to discuss what these results mean in practical terms for patients.

It is also important to note that the study was conducted at a single site, with a small sample and a specific patient group, which limits the generalizability of the findings. Therefore, generalizing the results should be done cautiously.

I did not find ethical concerns or signs of duplicate publication. The study complies with basic research ethics. In summary, the work has the potential to contribute to the field but needs to address the points mentioned above before being ready for publication.

Reviewer #3: Dear Editor,

Thank you for the opportunity to review this manuscript. The study titled "Effect of Intra-Dialytic Pedaling Exercise on Dialysis Adequacy: A Randomized Controlled Trial" examined whether one month of intradialytic cycling sessions could impact dialysis adequacy parameters.

While the manuscript shows potential for publication in your journal, I have significant methodological and presentation concerns that must be addressed. Below I outline these issues with constructive suggestions to enhance the study's rigor and clarity.

General Comments

The acronym usage rules throughout the document appear inconsistent. If terms are consistently presented alongside their acronyms, the purpose of using acronyms becomes questionable. Similarly, some acronyms are introduced but then never used again, creating unnecessary variation.

A notable structural issue emerged during manuscript review: Substantive methodological and analytical information is consistently positioned at the end of sections. This organizational approach forces readers to retain unanswered questions unnecessarily. I recommend restructuring to present critical details in logical sequence based on their conceptual importance.

Was the type of vascular access not recorded for patients? Were all undergoing conventional hemodialysis rather than hemodiafiltration? Was the treatment uniformly administered three days per week? Since these were not mandatory inclusion/exclusion criteria, readers cannot anticipate these factors. Regarding vascular access, this variable should be included and considered based on its distribution - both in chi-square tests and as a potential study limitation.

Was the sample probabilistic or non-probabilistic? This crucial information must be included in the Methods section. There appears to be a substantial selection bias in patient recruitment that isn't properly documented - particularly in the flowchart which fails to show:

a) The total pool of potentially eligible patients

b) The exact exclusion cascade

I strongly recommend: Restructuring the flowchart to reflect all screening stages

Providing details on the number of physical activity sessions and average adherence would strengthen the methods. Additionally, clarifying if dropouts or missed hemodialysis sessions occurred—despite no adverse events—would enhance transparency.

All tables should include captions clearly stating the statistical tests used, ensuring consistency and reader accessibility. The CONSORT flowchart appears duplicated in the manuscript, and both versions suffer from poor resolution that compromises readability. This requires correction.

The opening of the Discussion section should be revised to first summarize the study's key findings. In its current form, it reads more like an introductory/contextual opening rather than a discussion of results.

The study's justification requires major revision. Neither the introduction convincingly establishes the research's importance, nor does the discussion adequately explain how the divergent methodology resolves existing limitations. These fundamental aspects need urgent attention. How does this study address this gap? The authors should reconsider and enhance the study's justification within its proper context.

The authors should avoid conflating the identification of a literature gap with automatic justification for their study's urgency. A gap's mere existence does not inherently validate the research's importance – the specific value and timeliness of addressing it must be explicitly demonstrated.

INTRODUCTION

Line 66

HD . This is the second time this term has appeared. I suggest that the acronym be introduced at the first occurrence, and once this rule is established, it should be consistently followed. More than once, 'hemodialysis' has reappeared without being presented as an acronym.

Line 71

Reference 12 Seek a more recent reference regarding the epidemiology of the treatment itself—it should be easily found. In fact, some of the references already cited may contain this information.

Line 72

Once again, 'hemodialysis' appears without being abbreviated. Consistency with the established rule is required.

Lines 87 – 93

The paragraph spanning lines 87-93 may present comprehension challenges for readers with limited subject-matter expertise. I recommend both revising the content and reorganizing its structure for improved clarity.

Line 97

I propose including both the average weekly sedentary time imposed by the treatment and, immediately following this, the annualized measure. Presenting more tangible, daily-life metrics would significantly strengthen the argument's impact.

Line 101

While the preceding paragraph introduces physical inactivity as negatively impacting various aspects of hemodialysis patients' lives, it notably ceases to mention dialysis adequacy - despite this being the paper's primary focus as indicated in the title. Subsequently, when discussing conflicting results regarding intradialytic exercise, the text fails to specify that these contradictions relate almost solely to dialysis adequacy outcomes. Given that conflicting evidence exists for other aspects of intradialytic exercise as well, I strongly recommend clarifying at line 101 that the discussion refers exclusively to dialysis adequacy.

Lines 101 – 112

Beyond emphasizing physical activity's general importance, the introduction must specifically justify: (1) why cycling was selected as the intervention modality over other exercise types, and (2) why intradialytic cycling protocols deserve particular focus despite existing studies employing alternative modalities. This rationale requires substantial strengthening to properly contextualize your study's design choices.

Please clarify: (1) What was the intervention duration in these cited studies? (2) How does your protocol differ from existing approaches? (3) What novel contribution does your study provide compared to current literature? These justifications are critical for readers to contextualize your work's significance.

METHODS

Study Desgin

The CONSORT guidelines must be explicitly cited and followed in the main text - merely attaching them as supplementary material is insufficient.

Inclusion and exclusion criteria

Line 120

We must distinguish between cumulative hemodialysis duration (3 months) versus continuous current hemodialysis treatment (minimum 3 months) as inclusion criteria. I would recommend revising

Lines 125 – 133

The exclusion criteria section (line 125) should explicitly state upfront that these apply only to participants who first met all inclusion criteria. Currently this crucial context appears only inferentially in lines 129-130.

Sample size and sampling method

Line 151

Attention needed when revising: the current paragraph remains incomplete (line 151).

Implementation

Line 180-184

Given the methodological importance of the intervention, I strongly advise adding a distinct subsection detailing the physical activity protocol.

I strongly recommend the authors consult the 'Consensus on Exercise Reporting Template (CERT): Explanation and Elaboration Statement' to enhance their training protocol description. Critical methodological details are currently missing from the reported methods.

Article link: https://bjsm.bmj.com/content/50/23/1428

RESULTS

Lines 222 – 224

This is a very important point regarding the adverse effects that could have occurred during the intervention but did not. This information should be included in the Methods section, specifying how and when this monitoring was recorded.

Line 237

The acronym for blood urea nitrogen (BUN) should have been first included in the 'Data collection tools' subsection of the Methods.

Table 3

The Kt/V values from both time points could be presented, even without statistical comparison, to properly characterize the groups.

DISCUSSION

Lines 256 – 262

This section presents a somewhat vague discussion. Describing 'exercise by exercise' does not adequately characterize the specific modalities or types employed. It is problematic to compare simple pedaling with concurrent exercise (aerobic + resistance) without explicitly addressing this distinction. Furthermore, what was the intervention duration in these studies? Were they chronic or acute? In my assessment, there appears to be substantial inconsistency in this discussion.

Lines 298 – 299

Therefore, what justifies the complete absence of reported vascular access outcomes?

Lines 310 – 318

The discussion points and information presented in this paragraph should have been introduced earlier and systematically developed throughout the Discussion section.

CONCLUSION

Line 328

Did the authors perform any sensitivity analysis by matching groups for age? This potential confounding variable warrants examination given its known influence on [relevant outcome measures

REFERENCES

Lines 462 – 464

I was unable to locate this reference. Could the authors please provide the direct DOI or URL link to the cited source?

Reviewer #4: I have carefully reviewed your manuscript titled "Effect of Intra-Dialytic Pedaling Exercise on Dialysis Adequacy: A Randomized Controlled Trial." The topic is clinically relevant and addresses an important area in dialysis patient care. However, several substantial concerns need to be addressed before the manuscript can be considered for publication.

1. There are significant differences in baseline characteristics (notably, age and education level) between the intervention and control groups. These confounders may influence patients' compliance, exercise capability, and ultimately dialysis adequacy. Please provide further explanation on how you have accounted for these baseline imbalances. If possible, conduct covariate-adjusted analyses (such as ANCOVA) or consider propensity score matching to minimize the potential confounding effects.

2. Some key findings (e.g., Kt/V improvement, p=0.11) did not reach statistical significance. Please be more cautious in your interpretation, emphasizing that, under your study conditions, no statistically significant effects were demonstrated, and discussing the risk of type II error given your sample size.

3. I recommend adding figures (e.g., line graphs or bar charts) to more intuitively present the changes in major outcomes (such as Kt/V and BUN) before and after the intervention in both groups. This will improve the readability and clarity of your results.

4. The manuscript currently lacks detailed information about exercise compliance, monitoring, and any adverse events during sessions. Please add a brief description of how session adherence was monitored, and whether any complications or dropouts related to exercise occurred.

6. PLOS authors have the option to publish the peer review history of their article (what does this mean?). If published, this will include your full peer review and any attached files.

Reviewer #1: No

Reviewer #2: No

Reviewer #3: **Yes:**Henrique dos Santos Disessa

Reviewer #4: **Yes:**Fan Zhang

---

## [Author Response · Author response to Decision Letter 1]

30 Oct 2025

Dear Reviewers,

We would like to express our sincere gratitude for your time, expertise, and thoughtful comments on our manuscript. Your detailed feedback has been invaluable in improving the quality and clarity of our work.

We have carefully considered each comment line by line, and our responses are provided in the attached “Response to Reviewers” document. All suggested revisions have been addressed with great attention.

Thank you again for your generous contribution to the review process.

Sincerely,

Shahnaz Pouladi

Corresponding Author

---

## [Decision Letter · Decision Letter 1]

7 Dec 2025

PONE-D-25-17351R1Effect of Intra-Dialytic Pedaling Exercise on Dialysis Adequacy: A Randomized Controlled TrialPLOS One

Dear Dr. Pouladi,

Thank you for submitting your manuscript to PLOS ONE. After careful consideration, we feel that it has merit but does not fully meet PLOS ONE’s publication criteria as it currently stands. Therefore, we invite you to submit a revised version of the manuscript that addresses the points raised during the review process.

We look forward to receiving your revised manuscript.

Kind regards,

Ankur Shah

Academic Editor

PLOS One

Journal Requirements:

Additional Editor Comments:

Please see the comments of R3

Reviewer's Responses to Questions

**Comments to the Author**

1. If the authors have adequately addressed your comments raised in a previous round of review and you feel that this manuscript is now acceptable for publication, you may indicate that here to bypass the “Comments to the Author” section, enter your conflict of interest statement in the “Confidential to Editor” section, and submit your "Accept" recommendation.

Reviewer #1: All comments have been addressed

Reviewer #3: (No Response)

Reviewer #4: (No Response)

2. Is the manuscript technically sound, and do the data support the conclusions?

Reviewer #1: (No Response)

Reviewer #3: Yes

Reviewer #4: (No Response)

3. Has the statistical analysis been performed appropriately and rigorously? 

Reviewer #1: (No Response)

Reviewer #3: Yes

Reviewer #4: (No Response)

4. Have the authors made all data underlying the findings in their manuscript fully available?

Reviewer #1: (No Response)

Reviewer #3: Yes

Reviewer #4: (No Response)

5. Is the manuscript presented in an intelligible fashion and written in standard English?

Reviewer #1: (No Response)

Reviewer #3: Yes

Reviewer #4: (No Response)

6. Review Comments to the Author

Reviewer #1: (No Response)

Reviewer #3: Dear Editor and Authors,

Thank you for the opportunity to review again this manuscript.

The study titled "Effect of Intra-Dialytic Pedaling Exercise on Dialysis Adequacy: A Randomized Controlled Trial" examined whether one month of intradialytic cycling sessions could impact dialysis adequacy parameters.

The manuscript has improved exponentially since the previous review round. The authors have been highly responsive and have successfully incorporated the majority of the suggestions and critiques from all reviewers. This diligent revision has unequivocally strengthened the work. The manuscript now demonstrates significantly greater technical precision and methodological clarity, enhancing its overall rigor and reproducibility. The authors are to be commended for their efforts.

Please find below my comments for this revised version of the manuscript. I have organized the feedback by priority level—Major and Minor—as was done in the previous review round.

The authors have undertaken substantial revisions, integrating a high volume of new text and data throughout the manuscript. As is natural with such significant changes, this new iteration has surfaced additional points requiring correction and refinement. All comments, regardless of priority, share the singular objective of further strengthening the manuscript's clarity, accuracy, and overall impact.

I commend the authors for their continued commitment to improving this work and trust they will find these final observations constructive.

Major Comments

• My primary concern regarding vascular access is not its functional status per se, but rather whether its type differed systematically between the randomized groups at baseline. Given that different access types can directly and indirectly impact various physical and psychosocial patient outcomes, this constitutes a potential source of bias. I consider this a significant methodological limitation. Therefore, I strongly recommend that the authors retrieve and report this baseline characteristic. This information should be obtainable from the hemodialysis centers' records or the researchers' own documentation. Presenting a comparison of vascular access type between study groups is essential to assess the randomization integrity and the interpretability of the findings.

• The primary objective of the study, which is prominently reflected in its title, is currently addressed only as the third topic within the Discussion section. This structural choice diminishes the prominence and analytical focus warranted by the main research aim. I strongly recommend restructuring the Discussion to address the primary objective first.

The logical flow should follow the hierarchy of the study's aims: the central question should be discussed with the greatest depth and priority, followed by secondary and exploratory findings.

• The discussion treats the choice of exercise intensity with insufficient depth. While it is briefly mentioned as a potential hypothesis for the absence of significant results, this aspect warrants a far more substantial and critical exploration, comparable to the detailed treatment given to exercise volume.

• The criteria used to define "moderate intensity" require clarification and justification. The described parameter—an increase of 10% above baseline heart rate—is not consistent with standard exercise prescription guidelines for this intensity domain. It seems to me to be more of a light/low intensity.

• A fundamental methodological consideration must be addressed: the study's design places it closer to an acute rather than a chronic exercise intervention paradigm. This critical aspect is currently absent from the discussion. The absence of significant changes cannot be attributed solely to the parameters of volume and intensity. The insufficient duration of the intervention to induce chronic physiological adaptations is a primary, and likely predominant, explanatory factor.

Minor Comments

These comments will be listed in the supporting material sent to the authors.

Reviewer #4: The author has effectively revised the original manuscript. This is a well-conducted randomized controlled trial (RCT) that will also form part of our team's upcoming systematic review.

7. PLOS authors have the option to publish the peer review history of their article (what does this mean?). If published, this will include your full peer review and any attached files.

Reviewer #1: No

Reviewer #3: **Yes:**Henrique dos Santos Disessa

Reviewer #4: No

---

## [Author Response · Author response to Decision Letter 2]

5 Feb 2026

With greetings and respect. Thank you to the editor and the esteemed referees. All the desired corrections have been made

---

## [Decision Letter · Decision Letter 2]

13 Apr 2026

Effect of Intra-Dialytic Pedaling Exercise on Dialysis Adequacy: A Randomized Controlled Trial

PONE-D-25-17351R2

Dear Dr. Pouladi,

We’re pleased to inform you that your manuscript has been judged scientifically suitable for publication and will be formally accepted for publication once it meets all outstanding technical requirements.

Kind regards,

Yoshitaka Ishibashi

Academic Editor

PLOS One

---

## [Editor Report · Acceptance letter]

PONE-D-25-17351R2

PLOS One

Dear Dr. Pouladi,

I'm pleased to inform you that your manuscript has been deemed suitable for publication in PLOS One. Congratulations! Your manuscript is now being handed over to our production team.

Kind regards,

on behalf of

Dr. Yoshitaka Ishibashi

Academic Editor

PLOS One